# Glucocorticoid Receptor Agonists to Improve the Productivity and Health of Early-Weaned Pigs: What is the Best Method of Delivery?

**DOI:** 10.3390/ani10071124

**Published:** 2020-07-01

**Authors:** Hailey Wooten, Hwanhee Kim, Amanda R. Rakhshandeh, Anoosh Rakhshandeh

**Affiliations:** 1Department of Animal and Food Sciences, Texas Tech University, Lubbock, TX 79409, USA; hailey.wooten@ttu.edu (H.W.); hwanhee.kim@ttu.edu (H.K.); 2Department of Biology, South Plains College, Levelland, TX 79336, USA; arakhshandeh@southplainscollege.edu

**Keywords:** piglet, glucocorticoid receptor agonist, antibiotics, delivery method, growth performance

## Abstract

**Simple Summary:**

In previous studies, we demonstrated that injection of a glucocorticoid-like compound could effectively improve the productivity of early-weaned pigs and can be used as an alternative to antibiotics. In the current study, we explored the viability of other methods for delivering this glucocorticoid-like compound to newly weaned pigs, in the hopes of finding a delivery method that is effective, less labor intensive, and more animal welfare friendly. In this study, we compared two methods of glucocorticoid-like compound delivery, i.e., in-feed or in-water, to repeated intramuscular injections of the same compound. We also compared the effectiveness of the treatments with that of in-feed antibiotics. We found that the in-feed delivery method of the glucocorticoid-like compound had comparable positive effects on the growth performance of piglets with those of intramuscular injection of the glucocorticoid-like compound or antibiotics. Therefore, in-feed supplementation of the glucocorticoid-like compound is a suitable and viable alternative to the intramuscular injection or in-water delivery methods.

**Abstract:**

The purpose of the current study was to determine the best method of delivery for glucocorticoid receptor agonist (GRA) treatment. A total of 167 Pig Improvement Company (PIC) piglets (body weight (BW) 7.35 ± 1.24 kg) were weaned at 25.0 ± 0.81 days of age and randomly assigned to 14 treatment groups based on a 2 × 7 factorial arrangement with sex (gilts vs. barrows), in-feed antibiotic (ANT; 110 mg/kg in-feed tylosin), repeated intramuscular (I.M.) injection of GRA (two injections, 0.2 mg/kg BW dexamethasone (DEX)), low dose in-feed GRA (LF, 2.5 mg/kg diet DEX ), high dose in-feed GRA (HF, 5 mg/kg diet DEX), low dose in-water GRA (LW, 0.8 mg/L DEX ), high dose in-water GRA (HW, 1.6 mg/L DEX ), and no treatment control (CON) as the main factors. Body weight and feed intake were measured daily from days 0 to 7 and weekly from days 7 to 28 post-weaning. The interaction effect for average daily gain (ADG) was significant with gilts performing better in the I.M., ANT, and LF groups (*p* = 0.05). All treatment groups, with the exception of the HW group, had a higher ADG than the CON group. Gilts in the I.M., LF, and HF groups had the highest ADG compared to other treatment groups (*p* ≤ 0.05). Sex and the interaction between sex and treatments had no effect on the gain-to-feed ratio (G:F; *p* ≥ 0.21). All treatment groups had a higher G:F than the CON group (*p* ≥ 0.04). These results suggest that the low-dose, in-feed GRA treatment is the best GRA delivery method and is a suitable alternative to in-feed sub-therapeutic antibiotics.

## 1. Introduction

Early weaning exposes pigs to numerous stressors, which causes intestinal and systemic inflammation and impairs normal gut function and nutrient utilization post-weaning [1,2]. Ultimately, these effects lead to a significant drop in growth and a predisposition toward stress and disease later in life [3,4]. Common methods of improving post-weaning performance, such as the use of high-quality diets and sub-therapeutic antibiotics, have limitations because the former is expensive and the latter presents food safety issues. Therefore, alternative methods of mitigating the negative effects of weaning stress are needed. Previous studies in our laboratory have shown that administering a glucocorticoid receptor agonist (GRA) during early weaning can help reduce systemic inflammation and consequently improve post-weaning growth performance in pigs [2,5,6]. Dexamethasone was chosen as the GRA because its anti-inflammatory properties are well studied, it is frequently used in the livestock industry, and its short half-life eliminates the possibility of any residues at slaughter [2,7]. However, the GRA treatment in these studies was delivered via repeated intramuscular (I.M.) injection at two separate times. While this delivery method bypasses the first-pass intestinal metabolism, is effective, and is highly controllable, it can be stressful for piglets and labor intensive for producers. Thus, it may be impractical in commercial settings [8,9,10]. In comparison, treatments delivered orally would not cause stress and/or require more labor than is already used for the administration of in-feed antibiotics. Moreover, in monogastric animals, orally ingested GRA has been shown to have a bioavailability that is similar to that of I.M. injected GRA [11,12]. As such, the objective of the current study was to test and compare various common methods for delivering GRA treatment to pigs during early weaning.

## 2. Materials and Methods

The experimental protocol for this study was reviewed and approved by the institutional animal care and use committee of Texas Tech University (ACUC approval number 17022-02). This study was conducted at the Texas Tech University Swine Research Center (New Deal, TX, USA). Dexamethasone was used for the intramuscular injection (Phoenix Pharm Inc., Burlingame, CA, USA), in-feed (Patterson Veterinary Supply Inc., Devens, MA, USA), and in-water (Rising Pharmaceuticals Inc., Saddle Brook, NJ, USA) delivery of GRA. In-water and in-feed dexamethasone was added to the drinking water and the diet according to the manufacturer’s instructions, to ensure dexamethasone stability. Tylosin (Tylan 40, Elanco Inc., Greenfield, IN, USA) was used as the in-feed antibiotic.

### 2.1. Animals and Treatment

A total of 167 PIC piglets (84 gilts and 83 barrows, body weight (BW) 7.35 ± 1.24 kg; PIC USA, Hendersonville, TN, USA) were weaned at 25.0 ± 0.81 days of age and randomly assigned to 14 treatment groups based on a 2 × 7 factorial arrangement with sex (gilts vs. barrows), in-feed antibiotic (ANT; 110 mg/kg in-feed tylosin), repeated intramuscular (I.M.) injection of GRA (two injections 48 hours apart, 0.2 mg/kg BW dexamethasone (DEX)), low dose in-feed GRA (LF, 2.5 mg/kg diet DEX ), high dose in-feed GRA (HF, 5 mg/kg diet DEX), low dose in-water GRA (LW, 0.8 mg/L DEX ), high dose in-water GRA (HW, 1.6 mg/L DEX ), and a no treatment control (CON) as the main factors. Within the in-feed and in-water delivery methods, two levels of GRA inclusion were tested to account for any potential variability in oral consumption as well as the possible effects of first and second pass, i.e., intestinal and hepatic, metabolism of GRA, which might alter the effective dose of GRA. Anti-inflammatory doses (low dose) and immune suppressive doses (high dose) of GRA for in-feed and in-water use were determined based on the manufacturer’s recommendations. Once separated from sows, piglets were moved into a temperature-controlled nursery facility consisting of two rooms and housed in floor pens (3 or 4 pigs per pen, 3 pens per treatment, total of 42 pens, 12 gilts per treatment, 11 or 12 barrows per treatment). Pigs were housed in nursery pens with plastic slatted flooring and 1.5 times the minimum space was provided for each pig [13]. Male pigs were physically castrated during processing at 3 ± 1 days of age, and piglets were not creep-fed during the pre-weaning phase. Throughout the starter phase (days 0–28 post-weaning), pigs were fed ad libitum corn–soybean meal (SBM)-based diets according to a phase-feeding program (Table 1). The amino acid content of the diets was formulated to provide an optimum ratio to standardized ileal digestible (SID) lysine, according to the recommendations of Nutrient Requirements of Swine (NRC 2012) [14]. Pigs had free access to water, with pigs in the LW and HW groups having free access to water through mounted pen drinkers with a nipple (Trojan Specialty Products, Dodge City, KS, USA). Ice packs were routinely changed in the pen drinkers to keep the water temperature comparable to that of the on-site water drinking system.

### 2.2. Measurements

Measurements of growth performance, average daily gain (ADG), and average daily feed intake (ADFI) of each pen were taken weekly to determine the weekly G:F during the course of the study (i.e., 0–28 days post-weaning). Feed intake was measured daily for the first 7 days post-weaning and weekly thereafter. Daily water usage (disappearance) for the in-water treatment groups was calculated throughout days 0–7 post-weaning as the difference between the amounts of water (weight basis) provided and the water remaining in the drinker. Measurements were done at the exact same time every day to calculate the 24-h water usage for each pen. Drug consumption was approximated for in-feed and in-water treatment groups by evaluating the 24-h feed intake or water, respectfully.

### 2.3. Statistical Analysis

Data were analyzed using mixed procedures (PROC MIXED) in SAS (version 9.4, SAS Institute Cary, NC, USA). In a completely randomized factorial design, repeated measurement analysis of variance was used for data analysis over time. Normality and homogeneity of variances were confirmed using the univariate procedure (PROC UNIVARIATE). Outliers were determined as any value that differed from the treatment mean by ± 2 standard deviations. The fixed effects were treatment, sex, and day. Pen was the experimental unit and pen within room was used as the random effect. Body weight at weaning was used as a co-variate and, when appropriate (*p* > 0.10), the reduced model was used. Covariance structure was selected based on best fit according to Akaike information criterion (AIC) and Schwarz Bayesian information criterion (BIC). To evaluate the linear and quadratic effects of GRA intake on growth performance, a polynomial orthogonal contrast was tested for each sex group. Values were reported as least squares means ± SE. Treatment means were separated using the Tukey test. Significance was considered at *p* ≤ 0.05 and a tendency was considered at *p* ≤ 0.10.

## 3. Results

During the course of the study, pigs were free from major swine diseases and appeared to be healthy, only showing signs and symptoms of the stress of weaning. The analyzed nutrient contents of the diets were within 5% of the calculated values. No outliers were identified in normality and homogeneity of variance analysis. Growth performance results are presented in Table 2. The interaction effects between sex and treatments were significant: relative to CON gilts, ANT, I.M., and LF gilts had the highest ADG during the starter phase (*p* < 0.01), while ADG was highest in LF, I.M., and LW barrows, compared with CON barrows (*p* < 0.01). Overall, all treatment groups, with the exception of HW gilts and HF barrows, had a higher ADG than CON gilts and barrows during the starter phase (0–28 days post-weaning). In-feed ANT-treated gilts had a higher ADG than their barrow counterparts during the starter phase (*p* < 0.03). Only gilts receiving high-dose, in-water GRA treatments underperformed compared to their barrow counterparts in ADG (*p* < 0.04). There was no interaction between the main factors and time (day) for ADG (*p* ≥ 0.90). Overall, I.M., ANT, LF, HF, and LW pigs had better growth performance than the CON and HW pigs on days 7, 14, 21, and 28 post-weaning (*p* ≤ 0.05). No co-variate effect on ADG was observed (*p* = 0.56). Polynomial contrast analysis indicated that in-feed GRA increased the overall ADG in a quadratic fashion in both gilts and barrows (*p* < 0.01; Appendix A). Dietary supplementation above 2.5 mg/kg diet (LF) did not improve overall ADG (Appendix A). However, the contrast analysis suggested that when GRA is supplemented in water, it increased (*p* = 0.03) the overall ADG in gilts in a quadratic fashion, and tended to increase the overall ADG in a linear fashion in barrows (*p* = 0.07). Similar to the in-feed delivery method, GRA supplementation above 0.8 mg/liter of drinking water (LW) had no beneficial effects on the overall ADG of pigs (Appendix A). During the first week post-weaning, no relationship was observed between GRA intake, through water or feed, and the growth performance of the pigs (*p* ≥ 0.25; Appendix A).

The interaction between the main factors of sex and treatments (SEX × TRT) was significant for average daily feed intake (ADFI) during the starter phase (*p* = 0.01)—while ADFI was higher in gilts than in barrows for the CON and I.M. groups, they were not statistically different in other treatment groups. Barrows in the CON group had a higher ADFI than barrows in the I.M., ANT, LF, HF, and LW groups, but not HW barrows, during the starter phase (*p* ≤ 0.01). No interaction between the main factors and time (day) was observed for ADFI (*p* ≥ 0.36). Average daily feed intake consistently increased over the 4-week post-weaning period in all treatment groups (*p* < 0.01). The co-variate effect of body weight at weaning tended to be significant for ADFI (*p* = 0.09).

No interaction between the main factors of sex and treatments (SEX × TRT) was observed for the G:F ratio (*p* = 0.21). Additionally, the main factor of sex had no significant effect on the G:F ratio (*p* = 0.90). The main effect of treatment significantly affected the G:F ratio (Figure 1, *p* < 0.01)—while all treatment groups exhibited higher G:F ratios than the CON group, the groups with the highest G:F ratio, relative to control, were the ANT, I.M., LF, and LW groups (*p* < 0.01). No co-variate effect on G:F ratio was observed (*p* = 0.28).

## 4. Discussion

We and other workers have demonstrated that repeated I.M. injection of an anti-inflammatory dose of GRA improved the growth performance (ADG, BW, the G:F ratio) of early-weaned pigs during the starter phase by downregulating the measures of intestinal and systemic inflammation in pigs [2,5,6,15]. In those previous studies, I.M. injection was chosen as the preferred method of delivery because it allows the dose and timing of treatment to be accurately controlled. However, using this method is laborious, subjects pigs to additional stress and pain, and is likely not cost effective in a large commercial pig production setting [8,9,10]. Therefore, in the current study, we investigated alternative, less invasive and less labor-intensive methods (in-feed and in-water) for delivering GRA treatment to pigs during weaning. It is noteworthy that we have already explored the effects of GRA, in comparison to antibiotics, on measures of immune function, measures of digestive capacity and intestinal health, growth performance, and nutrient utilization (including nutrient digestibility) previously. The possible mechanisms through which GRA exerts its positive effects on animal production and well-being have also been discussed previously [2,5,6].

In the present study, the positive effects of GRA and ANT treatment on growth performance parameters were in general agreement with the previous findings in our laboratories and in the laboratories of others [2,5,6,15]. In the current study, the ADG of gilts outperformed that of barrows for all treatments, except for HW. This lower performance of barrows is likely associated with a predisposition of barrows toward inflammation and stress brought on by physical castration at a young age [16]. Indeed, barrows experience greater growth loss during weaning and have a higher cortisol response to stressors later in life when compared to gilts [17,18]. In contrast, the ADG of gilts was lower, as compared to barrows, in the HW groups. In the present study, we assumed that the water or feed intake reflected GRA consumption in the pigs that received GRA through water or feed. After accounting for oral GRA consumption, we found that the pigs that received the in-water GRA treatment consumed more GRA than the in-feed treatment groups. This difference may also, in part, explain why barrows in the HW group had a higher ADG than gilts, unlike for other treatment groups. Thus, these results suggest that the optimal dose of GRA for barrows, who are more sensitive to inflammation and stress during weaning, may not be the same as that for gilts and deserves further investigation.

In the current study, while all treatments, not including HW gilts, had positive effects on both growth and feed efficiency relative to the control, the treatments whose performance was the most comparable to ANT and I.M. were LF and HF. Gilts and barrows in both in-feed treatment groups had a high ADG that was similar to I.M. and ANT pigs. In addition, the effect of GRA on ADG in the I.M., LF, and HF groups was more consistent for both sexes than the effect of ANT during the starter phase. Finally, we did not observe a difference in feed efficiency between sexes throughout the course of the study. In addition, the G:F ratio was not different among ANT, I.M., LF, and LW pigs, suggesting that feed efficiency in pigs receiving low doses of GRA through feed or water is comparable to that of pigs receiving in-feed ANT. In the present study, the contrast analysis suggested that GRA supplementation above the anti-inflammatory dose (i.e., above LF and LW) did not have any beneficial effects on the growth of pigs during the starter phase. Indeed, previous studies have shown that higher doses of GRA can have negative effects on the growth and immunocompetence of pigs [2]. Finally, although no differences in growth performance were observed between the LW, I.M., and in-feed GRA treatments, we do not recommend in-water treatment, because the water intake of piglets can be influenced by various factors such as ambient temperature, diet composition, ingredient quality, health status, etc. [14]. Therefore, GRA consumption through water might become very variable and lead to inconsistent effects on pig performance and their ability to cope with stresses. Collectively, based on the observations of the current study, low-dose in-feed delivery of GRA to newly-weaned pigs is the more practical approach, considering that it is economically viable and more animal welfare friendly than other approaches, especially in large-scale operations. Thus, in-feed supplementation with GRA provides an alternative to in-feed antibiotic for newly-weaned pigs.

## 5. Conclusions

These results of the current study indicated that the lower dose, in-feed method is the most effective and animal welfare friendly method of delivering GRA to early-weaned pigs. Growth performance and feed efficiency in LF pigs rivaled that of I.M. and ANT treatments. Economically, this method is likely the most practical for a commercial setting, as it does not require the additional labor associated with administering injections. Finally, LF treatment does not pose the same animal and human safety risks associated with giving injections and using antibiotics, and thus it is an alternative to antibiotic uses.

## Figures and Tables

**Figure 1 animals-10-01124-f001:**
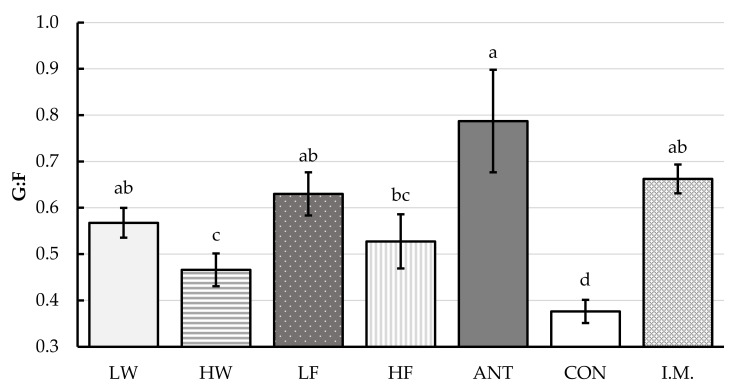
The impact of treatment, without regards to sex, on the gain-to-feed ratio (G:F). Values are least squares means (±largest SE) and represent the best estimate of mean based on repeated measurement analysis of variance. CON: control pigs received no treatment; I.M.: intramuscular injection pigs received 0.2 mg/kg BW of glucocorticoid receptor agonist (GRA) in the form of dexamethasone on days -1 and 3 post-weaning; ANT: antibiotic pigs received 110 mg in-feed tylosin/kg diet feed for the first week post-weaning; LF: low-dose in-feed GRA pigs received 2.5 mg GRA/kg diet for the first week post-weaning; HF: high-dose in-feed GRA pigs received 5 mg GRA/kg diet for the first week post-weaning; LW: low-dose in-water GRA pigs received 0.8 mg GRA/L water for the first week post-weaning; HW: high-dose in-water pigs received 1.6 mg GRA/L water for the first week post-weaning. ADG: average daily gain; ADFI: average daily feed intake; G:F: gain-to-feed ratio. ^a–d^ Means within a row lacking a common superscript letter are different (*p* < 0.05).

**Table 1 animals-10-01124-t001:** Ingredient composition and nutrient contents of the nursery diets.

Ingredients and Nutrient Contents	Diets ^2^		
	Phase 1	Phase 2	Phase 3
Ingredient, % (as fed)			
Corn	34.5	44.7	60.7
SBM	22	29.15	34.0
Whey powder	27.5	17.59	-
Plasma powder	4	3.02	-
Fish meal	4.5	-	-
Fat	2.0	1.5	1.0
Dicalcium phosphate	0.4	1.51	1.4
Calcium carbonate	0.7	0.7	0.7
Salt	0.45	0.35	0.25
Swine premix ^1^	4.0	3.0	2.0
Total	100	100	100
Calculated nutrient contents			
Metabolizable energy, MJ/kg	14.0	14.1	13.7
Crude protein (N × 6.25), g/kg	206	193	181
SID Lys ^3^, g/kg	13.6	12	10.1
Calcium, g/kg	8.2	8.6	7.4
STTD Phosphorus, g/kg	5.0	5.2	4.5
Calcium: STTD P	1.6	1.6	1.6

^1^ Providing the following amounts of vitamins and trace minerals (per kg of diet): vitamin A, 10075 IU; vitamin D3, 1100 IU; vitamin E, 83 IU; vitamin K (as menadione), 3.7 mg; D-pantothenic acid, 58.5 mg; riboflavin, 18.3 mg; choline, 2209.4 mg; folic acid, 2.2 mg; niacin, 73.1 mg; thiamin, 7.3 mg; pyridoxine, 7.3 mg; vitamin B12, 0.1 mg; D-biotin, 0.4; Cu, 12.6 mg; Fe, 100 mg; Mn, 66.8 mg; Zn, 138.4 mg; Se, 0.3 mg; I, 1.0 mg; S, 0.8 mg; Mg, 0.0622%; Na, 0.0004%; Cl, 0.0336%; Ca, 0.0634%, P, 0.003%; K, 0.0036%. SBM: soybean meal. ME: metabolizable energy. STTD: standardized total tract digestibility, ^2^ Phase 1, 2, and 3 were fed 0–7, 7–21, and 21–28 days post-weaning, ^3^ Lysine and other amino acids formulated on a standardized ileal digestible (SID) basis.

**Table 2 animals-10-01124-t002:** Impact of treatment and sex on various growth performance variables.^1^

Measurement	CON		I.M.		ANT		LF		HF		LW		HW		SE	*p* ≤		
	Gilt	Barrow	Gilt	Barrow	Gilt	Barrow	Gilt	Barrow	Gilt	Barrow	Gilt	Barrow	Gilt	Barrow		DM	Sex	DM × Sex
ADG kg/d																		
Overall	0.19 ^f,g^	0.17 ^g^	0.33 ^a^	0.28 ^a–d^	0.31 ^a,b^	0.25 ^c–e^	0.31 ^a–c^	0.28 ^a–d^	0.28 ^a–d^	0.24 ^d–f^	0.27 ^b–d^	0.27 ^b–d^	0.20 ^e–g^	0.27 ^c,d^	0.028	0.01	0.06	0.05
Day 7	0.07	0.05	0.10	0.11	0.10	0.11	0.06	0.08	0.07	0.05	0.07	0.13	0.07	0.05	0.055			
Day 14	0.14 ^b,c^	0.14 ^b,c^	0.25 ^a^	0.20 ^a,b^	0.23 ^a,b^	0.20 ^a,b^	0.25 ^a^	0.22 ^a,b^	0.17 ^a–c^	0.17 ^a–c^	0.16 ^a–c^	0.13 ^b,c^	0.09 ^c^	0.17 ^b,c^	0.050			
Day 21	0.26 ^c,d^	0.24 ^d^	0.41 ^a^	0.38 ^a,b^	0.43 ^a^	0.32 ^b,c^	0.42 ^a^	0.40 ^a^	0.43 ^a^	0.37 ^a,b^	0.39 ^a,b^	0.35 ^b,c^	0.31 ^c^	0.40 ^a^	0.036			
Day 28	0.28 ^d,e^	0.26 ^e^	0.55 ^a^	0.43 ^b^	0.50 ^a,b^	0.35 ^c–e^	0.50 ^ab^	0.43 ^b,c^	0.47 ^a,b^	0.38 ^b–d^	0.45 ^a–c^	0.48 ^a,b^	0.38 ^b–d^	0.44 ^a–c^	0.051			
ADFI kg/d																		
Overall	0.47 ^a–c^	0.53 ^a^	0.48 ^a,b^	0.42 ^c,d^	0.49 ^a–c^	0.37 ^c,d^	0.47 ^a–d^	0.45 ^b,c^	0.46 ^a–d^	0.47 ^b,c^	0.46 ^a–d^	0.45 ^b–d^	0.40 ^c,d^	0.45 ^a–c^	0.037	0.01	0.67	0.01
Day 7	0.17	0.19	0.15	0.16	0.19	0.11	0.18	0.16	0.16	0.17	0.17	0.18	0.17	0.18	0.060			
Day 14	0.30	0.35	0.31	0.29	0.25 ^a^	0.23	0.33	0.27	0.26	0.31	0.29	0.25	0.23	0.29	0.065			
Day 21	0.57	0.64	0.58	0.53	0.68	0.52	0.60	0.59	0.57	0.60	0.61	0.51	0.50	0.57	0.044			
Day 28	0.84 ^a–c^	0.94 ^a^	0.89 ^a,b^	0.72 ^d,e^	0.84 ^a–c^	0.63 ^e^	0.74 ^d^	0.78 ^b–d^	0.83 ^a–c^	0.78 ^b–d^	0.76 ^c,d^	0.86 ^a,bc^	0.72 ^d,e^	0.77 ^c,d^	0.055			
G:F																		
Overall	0.42	0.32	0.66	0.66	0.78	0.79	0.60	0.65	0.58	0.47	0.53	0.60	0.40	0.52	0.069	0.01	0.90	0.21
Day 7	0.41	0.26	0.55	0.68	0.46	1.15	0.32	0.56	0.40	0.27	0.32	0.75	0.11	0.27	0.310			
Day 14	0.47	0.39	0.78	0.68	1.42	0.84	0.72	0.81	0.59	0.53	0.55	0.43	0.36	0.57	0.349			
Day 21	0.48	0.37	0.70	0.70	0.63	0.61	0.69	0.68	0.76	0.62	0.64	0.66	0.61	0.70	0.253			
Day 28	0.33	0.27	0.61	0.58	0.59	0.56	0.69	0.55	0.56	0.48	0.58	0.55	0.52	0.56	0.229			

^1^ Values reported are least squares means (±largest SE) and represent the best estimate of mean based on repeated measurement analysis of variance. CON: control pigs received no treatment; I.M.: intramuscular injection pigs received 0.2 mg/kg BW of glucocorticoid receptor agonist (GRA) in the form of dexamethasone on days −1 and 3 post-weaning; ANT: antibiotic pigs received 110 mg in-feed tylosin/kg diet for the first week post-weaning; LF: low-dose in-feed GRA pigs received 2.5 mg GRA/kg diet for the first week post-weaning; HF: high-dose in-feed GRA pigs received 5 mg GRA/kg diet for the first week post-weaning; LW: low-dose in-water GRA pigs received 0.8 mg GRA/L water for the first week post-weaning; HW: high-dose in-water pigs received 1.6 mg GRA/L water for the first week post-weaning. ADG: average daily gain; ADFI: average daily feed intake; G:F: gain-to-feed ratio. DM: delivery method. ^a–g^ Means within a row lacking a common superscript letter are different (*p* < 0.05).

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
