# Peer review of "Glucocorticoid Receptor Agonists to Improve the Productivity and Health of Early-Weaned Pigs: What is the Best Method of Delivery?"

_animals, 2020, doi:10.3390/ani10071124_

Round 1

Reviewer 1 Report

Glucocorticoid receptor agonist (GRA): what is the best method of delivery?

Comments:Your articles are not suitable for publication in animals magazine. Although your research content is urgent in production and application, it is not substantial enough. There are several problems.

1. In lines 39, pig is changed to piglet and growth is changed to growth performance.

2. The author should consider the retention rate of glucocorticoid receptor agonists in feed and water, and suggest that the author measure the content of GRA after the addition of feed and water.

3. We need more actual measured values for feed formula. In addition, the ammonia content of several essential amino acids in nutritional ingredients should be supplemented as shown in table 1.

4. The experiment should further verify your results and suggest supplementary contents to enrich the research.

5. In table 2, the expression of p-value is not clear enough. In my opinion, it should be the delivery method, sex and the interaction between them. In addition, the author should supplement the p-value of each stage, which should not be omitted.

Author Response

Thank you for the comments and suggestions. We found the comments helpful and have modified the paper accordingly, as detailed below. We believe these changes result in an improved quality of the manuscript and hope they meet with your approval.

Reviewer 2 Report

A more detailed housing methodology is required, regarding barrows and gilts.

How many barrows and gilts were used per treatment?

Were they housed separately? if Yes, how many pens per sex?

Author Response

Thank you for the comments and suggestions. We found the comments helpful and have modified the paper accordingly. We believe these changes result in an improved quality of the manuscript and hope they meet with your approval.

Reviewer 3 Report

Dear Autors,

Dear Authors, the work provides important information on the route of GRA administration and its effectiveness. This is important for improving animal welfare and the production effects achieved. In the whole work, I would like to pay attention to a few details which I put below:

The authors write, glucocorticoids give similar results as antibiotics. Antibiotics counteract the development of bacterial pathogens, while GRA inhibits inflammatory processes. I am not convinced that this comparison is not too far-reaching.

In lines 42–43, the authors write that stress causes enteritis in pigs. It would be more accurate to use the statement that weaning piglets is a stressful disorder resulting in increased susceptibility to disease.

Authors should include some information on the limitations of the proposed method, the possible side effects of using GRA in animals.

The methodology should include the producer of the GRA used in these research.

I suggest putting the sentence from line 163-166 in the introduction chapter.

In items 16 and 17 of literature there is no year of publication.

Best regards

Author Response

(The authors gave the same response as above.)

Reviewer 4 Report

The paper is well written and the content is interesting to the scientific community. I recommend publication of the manuscript.

There are a few points that the authors should consider:

  • Outliers. Please indicate if any outliers were detected and whether there was any non-statistical reason to discard them.
  • Please explain in which cases initial body weight was used as a covariable, and whether it was significant.
  • GRA was administered in water and feed at two different levels. In order to test whether the dose effect was significant, it could be interesting a set of orthogonal polynomials to test for linear and quadratic effects of level of GRA including the negative control and the treatments containing GRA.
  • the authors mention that there was no response in performance to intake of GRA. One would expect that if GRA has an effect, the effect should change with the dose. Have the authors considered that the levels used in water and feed were excessive and that this was the reason for not finding a response to dose?
  • Performance of the piglets in the negative control was rather poor, particularly in G/F. How was this performance compared to other studies in the same farm? Is this low performance normal. One would expect G/F ratios of  0.6  or higher in this stage of growth

Author Response

(The authors gave the same response as above.)

Round 2

Reviewer 2 Report

Can be accepted in its current form

Author Response

This reviewer does not have any suggestion for change. However, we agreed with the editor's suggestion for providing a more descriptive title.  We have changed the title accordingly:
"Glucocorticoid receptor agonists to improve the productivity and health of early-weaned pigs: what is the best method of delivery?"